# Effect of Rosmarinic Acid and Ionizing Radiation on Glutathione in Melanoma B16F10 Cells: A Translational Opportunity

**DOI:** 10.3390/antiox9121291

**Published:** 2020-12-16

**Authors:** Amparo Olivares, Miguel Alcaraz-Saura, Daniel Gyingiri Achel, Miguel Alcaraz

**Affiliations:** 1Radiology and Physical Medicine Department, School of Medicine, University of Murcia, 30100 Murcia, Spain; amparo.o.r@um.es (A.O.); Miguel.Alcaraz@um.es (M.A.-S.); 2Applied Radiation Biology Centre, Radiological and Medical Sciences Research Institute, Ghana Atomic Energy Commission, Legon-Accra GE-257-0465, Ghana; d.achel@gaecgh.org

**Keywords:** radiation effects, radiosensitizers, glutathione, melanoma, PNT2, B16F10

## Abstract

To explain a paradoxical radiosensitizing effect of rosmarinic acid (RA) on the melanoma B16F10 cells, we analyzed the glutathione (GSH) intracellular production on this cell (traditionally considered radioresistant) in comparison with human prostate epithelial cells (PNT2) (considered to be radiosensitive). In PNT2 cells, the administration of RA increased the total GSH content during the first 3 h (*p* < 0.01) as well as increased the GSH/oxidized glutathione (GSSG) ratio in all irradiated cultures during all periods studied (1h and 3h) (*p* < 0.001), portraying an increase in the radioprotective capacity. However, in B16F10 cells, administration of RA had no effect on the total intracellular GSH levels, decreasing the GSH/GSSG ratio (*p* < 0.01); in addition, it caused a significant reduction in the GSH/GSSG ratio in irradiated cells (*p* < 0.001), an expression of radioinduced cell damage. In B16F10 cells, the administration of RA possibly activates the metabolic pathway of eumelanin synthesis that would consume intracellular GSH, thereby reducing its possible use as a protector against oxidative stress. The administration of this type of substance during radiotherapy could potentially protect healthy cells for which RA is a powerful radioprotector, and at the same time, cause significant damage to melanoma cells for which it could act as a radiosensitive agent.

## 1. Introduction

Rosmarinic acid (RA) is a common ester derived from caffeic acid and (R)-(+)-3-(3,4-dihydroxyphenyl) lactic acid present in many plant species. RA shows numerous biological activities of medical interest, including anti-viral, anti-bacterial, anti-inflammatory and antioxidant effects [1,2,3].

RA has been shown to have significant antigenotoxic and radioprotective capacities in different normal and tumor cell lines against damage induced by ionizing radiation (X-rays and gamma radiation) and by ultraviolet radiation [4,5,6,7,8,9,10]. However, it has also been shown to paradoxically increase radioinduced cellular damage in B16F10 metastatic melanoma cells, and thus acts as a radiosensitizing agent [11].

Cellular radioresistance in numerous healthy and tumor cell lines is mediated by increased glutathione (GSH) synthesis that leads to the elimination of ROS produced both by exposure to ionizing radiation and chemotherapy [12,13]. Intracellular GSH is the main thiol component that acts to combat oxidative stress in normal tissues. In cancerous tissues, different studies suggest that an increase in GSH levels is involved in cellular resistance to radiotherapy and chemotherapy, while the depletion of endogenous GSH levels is thought to increase the efficacy of treatments with ionizing radiation or chemotherapeutic agents [14,15,16,17,18,19]. However, numerous authors suggest that the increase or reduction in intracellular GSH levels caused either by activating or blocking its synthetic mechanisms affect both normal and tumor cells, making these selective treatment modalities ineffective in cancer patients [20].

In this study, we analyzed the radioprotective effect of RA on intracellular GSH levels in cells traditionally considered radiosensitive (human prostate epithelial cells) and the effect produced by RA on the availability of GSH in other cells considered radiosensitive (B16F10 melanoma cells) which could justify this paradoxical radiosensitizing effect. Potentially, the simultaneous administration of this type of substances along with radiation at the same time could protect healthy cells, while allowing significant damage to melanoma cells; these findings suggest the possibility of incorporating this hitherto impossible to perform treatment strategy to patients undergoing radiation therapy.

## 2. Materials and Methods

### 2.1. Chemicals and Reagents

Rosmarinic acid (RA) (Figure 1), was obtained from Extrasynthese S.A. (Genay, France). RPMI 1640, Ham’s F10, streptomycin, penicillin, phosphate-buffered saline (PBS), Bovine serum albumin (BSA fraction V), and Fetal bovine serum (FBS) were obtained from Gibco (Life Technologies S.A., Madrid, Spain); glacial acetic acid and ethanol were obtained from Scharlao SL (Madrid, Spain).

### 2.2. Cell Lines and Culture Conditions

In this study, two cell types selected based on their radiosensitivity status were used: cells traditionally considered radiosensitive (human prostate epithelial cells (PNT2)); and B16F10 cells which are traditionally considered to be very radioresistant [13]. The normal epithelium prostatic cell line (PNT2) was obtained from the European Collection of Cell Cultures (ECACC, Salisbury, UK), Health Protection Agency, Culture Collection (catalogue no.:95012613, Salisbury, UK). The PNT2 cells were cultivated in RPMI 1640 (Sigma Aldrich) supplemented with 10% FBS, 2 mM glutamine, streptomycin and penicillin (100 μg/mL and 100 UI/mL, respectively). The mouse metastatic melanoma cell (B16F10) line was kindly provided by Dr. V. Hearning (NIH, Bethesda, MA, USA), and cultured in Dulbecco’s Modified Eagle’s medium (DMEM)/F12K (1:1), (Sigma-Aldrich St. Louis: USA) supplemented with 10% FBS (Gibco, BRL, Louisville, KY, USA), 4 mM L-glutamine, penicillin (100 IU/mL) and streptomycin (100 µg/mL).The cultures were maintained at 37 °C, a relative humidity of 90–95% and an atmosphere of 5% (PNT2) and 7.5% (B16F10) CO2. Tests were carried out to confirm the absence of Mycoplasma spp. throughout the study.

To analyze the effect of RA on glutathione levels in PNT2 and B16F10 cell lines, cell culture experiments were performed as previously described [11]. Briefly, the cultures were incubated in 200 µl growth medium and allowed to adhere for 24 h after cell seeding in both types of assays to get cells adapted to the culture conditions and to adhere to the bottom of the wells. For the PNT2 cells, 3.200 cells/wells and for B10F16, 2.500 cells/well were established as optimal cell seeding concentrations.

After 48 h of cell incubation, the culture medium was changed to 200 µl of fresh medium (control cultures) to which 25 µl of RA dissolved in phosphate-buffered saline (PBS) at a concentration of 30 µM were added (cultures treated with RA). Immediately afterwards, cells were exposed to 20 Gy of X-rays followed by the determination of GSH concentrations after 1h or 3h according to the period studied (irradiated control cultures and cultures treated with RA and irradiated).

### 2.3. Irradiation

An Andrex SMART 200E (Yxlon International, Hamburg, Germany) X-ray producing equipment with the following characteristics was used: 200 kV, 4.5 mA, dose rate of 1.3 cGy/s at a focus-object distance (FOD) of 35 cm and a total dose of 20 Gy. The doses of radiation administered were continuously monitored inside the X-ray cabin by means of UNIDOS^®^ Universal Dosimeter with PTW Farme^®^ ionization chambers TW30010 (PTW-Freiburg, Freiburg, Germany) and the final radiation dose was confirmed utilizing thermoluminescent dosimeters (TLDs) (GR-200; Conqueror Electronics Technology Co Ltd., Shenzhen, China).

### 2.4. GSH Assay

The GSH/oxidized glutathione (GSSG)-GloTM Assay (Promega, Madison, MI, USA) was used to determine and quantify GSH level in PNT2 and B16F10 cell lines. These assays are a luminescence-based system to detect and quantify total glutathione (GSH + GSSG), GSSG and GSH/GSSG ratios in cultured cells. The assay provides a simple multiwell-plate format where stable luminescent signals are correlated with the GSH or GSSG concentration of a sample. Light from luciferase depends on the amount of luciferin formed, which in turn depends on the amount of GSH present. Thus, the luminescent signal is proportional to the amount of GSH. In each well of a 96-well microtiter plate, 200 microliters of complete medium and cells was seeded. Cells were treated with test solutions of rosmarinic acid (25 µL). Cell density was determined and corrected by Bradford’s analysis [21]. The method was carried out according to the manufacturer’s instructions. The light emitted in the presence of GSH was quantitated in relative light units (RLU). The intensity of emitted light quants was directly related to GSH content in the tested sample. Cells were trypsinized and their fluorescence intensity was analyzed using FLUOstar^®^ Omega (BMG Labtech, Offenburg, Germany). All experiments were repeated eight times.

### 2.5. Statistical Analysis

Analysis of variance complemented by a contrast of means to determine the degree of dependence and correlation between the variables was performed and further complemented with regression and linear correlation analysis amongst the quantitative variables. Significant *p* values of less than 0.01 (*p* < 0.01) were deemed significant.

## 3. Results

The standard curve for total GSH assay shows good linearity up to a concentration of 8 µM, with some degree of saturation at higher concentrations. Given that the results obtained do not exceed 500,000 RLUs, they lie within the linear area of the dose–response curve, presenting a suitable slope and reproducibility for the study (Figure 2).

In the control cultures (C) statistically significant differences in the concentrations of total GSH were determined between the two cell lines studied (*p* < 0.001); during the two post-irradiation periods studied (1h and 3h) a greater amount of total GSH was determined in the B16F10 melanoma cells which was double the total GSH concentration determined in the PNT2 cells (Figure 3a).

Exposure to 20 Gy of X-rays (Ci) produced a significant increase in total GSH in the two periods studied and in both cell lines (*p* < 0.001), which could be interpreted as an immediate cellular defense mechanism against X-ray-induced cytotoxic damage (Figure 3a). In percentage terms, the increase in total glutathione concentration was greater in the PNT2 cells (90%) than that in B16F10 melanoma cells (60%) when compared to their non-irradiated control counterparts (*p* < 0.001) (Figure 3b).

The administration of RA dissolved in PBS did not produce statistically significant differences in the amount of total GSH concentration with respect to its controls in either of the two cell lines.

In the irradiated cells, the GSH/GSSG ratio increased significantly compared to that in the non-irradiated control cells (*p* < 0.001) during the two post-irradiation periods studied in both cell lines, with both cell lines showing a similar curve trend (*p* < 0.001) (Figure 4a). This connotes a significant increase in the endogenous GSH/GSSG ratio caused by exposure to X-rays.

The administration of RA dissolved in PBS to non-irradiated cell cultures produced different effects on the GSH/GSSGH ratios according to the cell line studied (Figure 4b). In PNT2 cells, the administration of RA produced an increase in the GSH/GSSG ratio during all the periods studied (1h and 3h), although it only reached statistical significance after 3h of its administration (*p* < 0.01). On the contrary, in the B16F10 cells, the administration of RA produced a significant decrease in the GSH/GSSG ratio 1 h after its administration (*p* < 0.01), reaching values similar to the ratio determined in non-irradiated control cells at 3h. (Figure 4b).

Exposure of cells to 20 Gy of X-rays previously treated with RA dissolved in PBS also showed a different response in each of the cell types. Figure 4c shows the GSH/GSSG ratio in irradiated cells treated with RA. In the PNT2 cells, the GSH/GSSG ratio increased significantly after exposure to X-rays (*p* < 0.001), expressing an increase in the amount of reduced glutathione available in the cells to reduce the cytotoxic damage induced by X-rays as a consequence of presence of RA administered before exposure to X-ray. In melanoma B10F16 cells, significantly lower levels in terms of GSH/GSSG ratio were also obtained after irradiation compared to non-irradiated RA-treated melanoma cells (*p* < 0.01) (Figure 4c). Surprisingly, the GSH/GSSG ratios observed in irradiated cells treated with RA remained at the same level as the non-irradiated B16F10 control cells (Figure 4a). This shows that the increase in the GSH/GSSG ratio in the irradiated RA-treated melanoma cells does not represent a real increase with respect to the ratios observed in the control cells (untreated and non-irradiated controls).

Figure 4d shows the GSH/GSSG ratio of RA-treated irradiated cells compared with the irradiated control cells. In PNT2 cells where exposure to X-rays produced a significant increase in the GSH/GSSG ratio, the administration of RA produced a significant increase (*p* < 0.001) in cells that were previously treated with RA; which demonstrates some degree of additivity or synergy of RA to the increased GSH/GSSG ratio. This might be considered as an enhanced capacity to eliminate free radicals induced by X-rays.

In contrast, when the B16F10 melanoma cells were exposed to X-rays, a significant increase in the GSH/GSSG ratio was observed, however, upon the administration of RA a statistically significant decrease in the GSH/GSSG ratio (*p* < 0.001) was elaborated. This observation expresses diminished protective capacity in cells previously treated with RA and could be interpreted as an increased oxidative stress and thus may explain the higher radiation-induced cellular damage in these cells.

In short, in B16F10 melanoma cells, the administration of RA and its subsequent exposure to 20 Gy of X-rays showed a GSH/GSSG ratio similar to that of control B16F10 melanoma cells; this abolished the increased GSH/GSSG ratio observed in the irradiated control cells and would explain the decrease in their radioprotective capacity against the damage induced by the 20 Gy of X-rays administered.

Figure 5 shows that the administration of RA to irradiated B16F10 melanoma cells abolishes the increase in the GSH/GSSG ratio determined in the irradiated non-RA-treated B16F10 melanoma cells and producing GSH/GSSG ratios similar to those of the control cells; which could portray a high oxidative stress in the cells and explain the greater damaging effect of X-rays in these B16F10 melanoma cells. On the contrary, in PNT2 cells, the administration of RA produced a GSH/GSSG ratio which was significantly (*p* < 0.001) higher than that produced in irradiated non-RA-treated PNT2 cells. This observation could be explained by the free radical scavenging effect of RA on free radicals produced by radiation. This action of RA eliminates the free radicals in a timely manner, protecting the oxidation of endogenous cellular glutathione and consequently leading to an increase in the GSH/GSSG ratio. All this would express a greater protection capacity against oxidative stress induced by X-rays (Figure 5).

## 4. Discussion

Glutathione, and its precursor cysteine (Cys), are sulfhydryl (thiol) compounds that constitute major intracellular antioxidants with the ability to eliminate ionizing radiation-induced oxidative stress. On the outside of the cell, Cys is found in its oxidized form, Cystine, which must be reduced to gamma-glutamylcysteine to be transported through the cell membrane by GSH reductase and released inside the cell where it becomes a precursor to GSH [22,23,24].

In melanocytes, Cys has an additional destination since it is closely associated with one of the pathways of melanogenesis to form pheomelanine [22,25]. The intracellular concentration of Cys is considered the main regulatory mechanism for the formation of another type of melanin. The higher the intracellular Cys and GSH concentration, the greater the activation of the pheomelanine synthetic pathway and the lower the eumelanin synthesis [18,23,26]

It has been demonstrated that exposure to 10 Gy of X-rays on PNT2 and B16F10 cells produces a decrease in cell survival of 29% and 42%, respectively, after 48h of incubation [11,27,28]. Given these results, it could be thought that cell death produced by radiation-induced oxidative stress should show a depletion of intracellular GSH and a decrease in the GSH/GSSG ratio as a consequence of radiation-induced oxidative stress [11,28]. However, during the first three hours after exposure to ionizing radiation, our results in both cell lines were different (a significant increase in total GSH and in the GSH/GSSG ratios showing a similar pattern that can be interpreted as a reactive cellular response). References describing this initial response in these cell lines are unavailable, although some authors have described, using lower doses of ionizing radiation, increases in total GSH and in the GSH/GSSG ratios in blood and in different rat organs during the first hours after exposure to irradiation. These observations have been interpreted as an intracellular reactive response to eliminate free radicals and which has even been considered as a cellular or adaptive response to exposure to ionizing radiation in organisms [29,30].

The administration of RA to PNT2 cells at the concentrations evaluated does not present cellular toxicity and provides a protection factor against genotoxic damage induced by radiation of 50%; in addition to a 30% reduction in ionizing radiation-induced cell death after 48h of cell incubation [11]. This genoprotective and radioprotective capacities expressed by RA is well described on different cell types in vitro or in vivo [4,5,6,7,8,9,10] and even against damage induced by ultraviolet radiation [6,31].

This radioprotective capacity of RA has been attributed to its chemical structure. It is known that the capacity of RA to scavenge hydroxyl radicals is based fundamentally on the combination of a ring system with conjugated double bonds in the polyphenolic skeleton, mainly the o-dihydroy-phenol or catechol structure; for this reason, the presence of two catechol groups in the RA conjugated to a carboxylic acid functional group increases its antioxidant activity in an aqueous medium. In fact, based on these structural considerations, the administration of RA before irradiation is consistent with its antioxidant and free radical scavenging activities [6].

We found no references on the effect of RA on intracellular GSH in the PNT2 cells. In brain, kidney and liver cells treated in vitro with high concentrations of RA, the inhibition of the enzymes glutathione reductase (GR) and glucose 6-phosphate dehydrogenase (G6PD) have been described, highlighting that, although RA has antioxidant properties, high concentrations of RA inhibit both G6PD and GR activity. GR inhibition may influence total intracellular GSH levels and lowers cellular defense against oxidative stress. They indicate that the mechanism RA inhibition may be due to the interaction of RA with sensitive SH groups present in G6PD. G6PD activity is thought to depend on the intactness of SH groups; the inhibitory effect of RA may be due to binding thiol groups in the enzyme. In consideration of the amino acid residue in the substrate and NADP+-binding site or catalytic site, cysteine residues were found, which play an essential role for the activation of G6PD. Finally, they suggest that because G6PD is an important enzyme in many reductive biosynthesis and GSH regeneration, its inhibition by RA may induce oxidative damage that causes hemolytic anemia in G6PD-deficient individuals [32]. However, after exposure to X-rays, there was an increase in total GSH concentration with the GSH/GSSG ratio not only increasing above the control values, but also over the reactive increase produced by ionizing radiation in PNT2 cells which were not treated with RA. This enormous increase produced by RA could be interpreted as a scavenging effect of radioinduced free radicals that would protect the oxidation of endogenous GSH and would show an additive or synergistic effect of RA with intracellular GSH against oxidative stress induced by ionizing radiation. These increases in endogenous reduced GSH would explain these radioprotective effects of RA [4,5,6,7,8,9,10] in agreement with the effects described with other cytoprotective agents [14,15,16,17,18]. A situation that has also been described in the use of other different chemical and physical agents [33,34].

In B16F10 melanoma cells, the administration of RA at the concentrations tested does not show cellular toxicity either. However, in the irradiated cells, RA is shown as a radiosensitizing agent producing a significant increase in cell death that reaches 73% compared to the irradiated control cells. This radiosensitizing capacity of RA has been described only in melanoma cells [11] although it is also produced by some other agents such as Carnosol [35].

We have not found previous references on the effect that RA produces on intracellular GSH in B16F10 melanoma cells. In our study, the administration of RA to B16F10 cells decreased the GSH/GSSG ratio with normal concentrations of total GSH, which shows the significant increase in oxidized glutathione produced by the administration of RA. Furthermore, after exposure to X-rays, the administration of RA abolishes the increase in the GSH/GSSG ratio expected in irradiated B16F10 cells, maintaining it at significantly levels lower than that of the control cells. This situation involves a blockage of endogenous GSH available in the cell and a reduction in the reduced glutathione available to eliminate radiation-induced free radicals. In short, a restrictive effect of RA is observed on the availability of endogenous GSH that would allow a higher damaging capacity of free radicals induced by ionizing radiation.

We previously attributed this paradoxical radiosensitizing effect of RA, which was observed only in melanoma cells, to melanogenesis, as this is a specific activity of this kind of cell line which differentiates it from other cells lines wherein RA acts as a powerful radioprotective agent [11,35].

In agreement with this hypothesis, it has been previously described that RA stimulates melanogenesis by increasing the content of melanin and the expression and activation of thyrosinase in mouse B16 melanoma cells after 48h of stimulation [6,36] probably through the activation of protein kinase A [37,38,39].

Literature regarding the influence of RA on melanogenesis is not available [6]. However, caffeic acid (of which RA is a dimer) has been described and considered as a structure analogous to RA [40] that activates melanogenesis leading to intracellular depletion of GSH with increase in ROS levels in B16F10 melanoma cells. In addition, in this process, a concomitant decrease in the activity of alanine aminotransferase and an increase in the level of malondialdeheide are observed, which reflects a decrease in the lipid peroxidation capacity and a decrease in the free thiol content in different organs (liver and kidney) [37,39]. All these results show that the activation of melanogenesis and the production of melanin is carried out through the pheomelanin formation pathway

The most significant structural difference between the two types of melanin is the presence of an aromatic ring containing sulfur in pheomelanin which is absent in eumelanin. Pheomelanine incorporates cysteine in its structure. An important cellular reserve of cysteine is glutathione, considered the most important cellular antioxidant. Activation of pheomelanine synthesis can lead to the deprivation or depletion of intracellular GSH stores that would no longer be fully available to eliminate radiation-induced free radicals [41,42,43]. In this sense, it has been described that the use of other antioxidants with radioprotective capacities such as Apigenin, DMSO, diphosphonates, and *Pycnanthus angolensis* extracts maintain their radioprotective capacities in B16F10 cells, possibly because they have no impact on melanogenesis [8,10,25,26,27].

Furthermore, it has been suggested that pheomelanine can generate ROS and directly increase cell damage. Although it has been suggested that this generation of ROS can occur in the absence of any type of exposure to electromagnetic radiation [43,44], it has been described that exposure to UVA rays [45], stimulation by visible light in the presence of Zinc [46] or ionizing radiation [47,48] increase the generation of ROS by affecting pheomelanin formation.

In our study, the observed paradoxical radiosensitizing effect of RA on B10F10 melanoma cells could be due to the sum of different factors. The activation of melanogenesis through the pheomelanin pathway could cause a decrease in intracellular GSH levels making it unavailable to scavenge radiation-induced ROS. Synthesis of pheomelanine is accompanied by an increase in malondialdhehyde levels and a decrease in lipid peroxidation production adding to the pool of ROS induced by ionizing radiation. Furthermore, part of the RA used in the activation of pheomelanine would not be available as an antioxidant in the aqueous cytoplasmic medium to scavenge radioinduced free radicals. Finally, pheomelanin interacting with ionizing radiation can directly generate ROS that would add to the radiation-induced free radicals.

The situation of reduced GSH deprivation could also be affected by other known factors such as a lower superoxide dismutase activity (capable of scavenging superoxide radicals) of the highly metastatic cell line B16F10 which is smaller than that observed in other metastatic melanoma lines [38,49]; and even the inhibitory effect of RA on glutathione reductase and glucose-6-phosphate dehydrogenase [32] and glutathione S-transferase [50], that reduces intracellular NADPH concentrations which also reduces the capacity of the cell to convert oxidized glutathione (GSSG) to reduced glutathione (GSH).

Some clinical and experimental data could support these radiosensitization mechanisms shown by the administration of RA in melanoma cells. Melanoma has been considered a radioresistant cancer, although the multiple underlying mechanisms are not sufficiently clarified [18]. In melanoma and in some other tumor cell lines, radioresistance is mediated by the greater presence or capacity for synthesis of GSH that leads to the elimination of ROS produced by irradiation [12,13]. However, significant evidence has accumulated that melanomas have a wide range of sensitivities to radiation. [51,52]. Pigmented melanoma cells show higher cell survival against ionizing radiation; as a function of the increased levels of melanin (evaluated by the increase in tyrosinase activity and therefore eumelanin) against exposure of 15 Gy of radiation showing a greater cellular capacity of radiation resistance. Similarly, experimental and clinical results indicate that the inhibition of melanogenesis (as in amelanotic melanomas or with an increase in pheomelanin) could be used for the radiosensitization of melanoma cells and improve the efficacy of radiotherapy in the treatment of melanomas. Some authors have even pointed out that the presence or lack of intense melanin pigmentation should be taken into account when choosing therapeutic options, since experimental and clinical data suggest that melanin could be a prognostic factor for the radiosensitization of melanoma cells [52]. Interestingly, the same approach can potentially sensitize melanoma cells not only to ionizing radiation but also to chemo- or immuno-therapy [51,52,53]. In addition, it has also been suggested that Melanin, although believed to be very stable, undergoes degradation in the organism, probably with the participation of NADPH-oxidoreductases (NOX), and upon exposure to visible light, ultraviolet radiation. [53,54,55] and ionizing radiation [47] Substances adsorbed during melanogenesis, some of which are very active (e.g., Fe) may be released back during the process of melanin degradation. This process is considered to be responsible for the toxicity and phototoxicity of melanin, which is stronger and more intense for pheomelanin than for eumelanin [52,56,57,58].

Obviously, more studies are necessary to confirm our results and to better analyze the antioxidant defense response and its interaction with radioinduced ROS (using DPPH assay, Ferric reducing antioxidant power assay, Nitro blue terazolium reduction assay, or ROS assay), and also by determining the capacity of detoxifying enzymes like SOD and glutathione peroxidases (GPx).

## 5. Conclusions

Selective modulation of GSH synthesis could be an effective method to achieve radiosensitization of tumor cells and/or radioprotection of healthy cells against toxic agents that induce free radicals. An increase in the levels of GSH or in the capacity of its synthesis by normal cells, would increase its cellular radio resistance, with the consequent radioprotective effect. On the contrary, a decrease in the availability of GSH or in the cellular capacity for its synthesis would increase its radiosensitivity to the effects of ionizing radiation or certain chemotherapeutic substances. Studies on substances like rosmarinic acid could help clarify mechanisms allowing protection of healthy normal cells while exclusively injuring neoplastic cells. Potentially, the simultaneous administration of this type of substance along with radiation at the same time could provide protection to healthy cells, while allowing significant damage to melanoma cells; these findings suggest the possibility of incorporating this hitherto impossible to perform treatment strategy to patients undergoing radiation therapy.

## Figures and Tables

**Figure 1 antioxidants-09-01291-f001:**
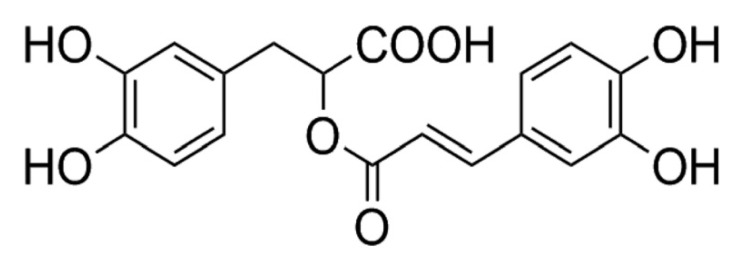
Chemical structure of rosmarinic acid.

**Figure 2 antioxidants-09-01291-f002:**
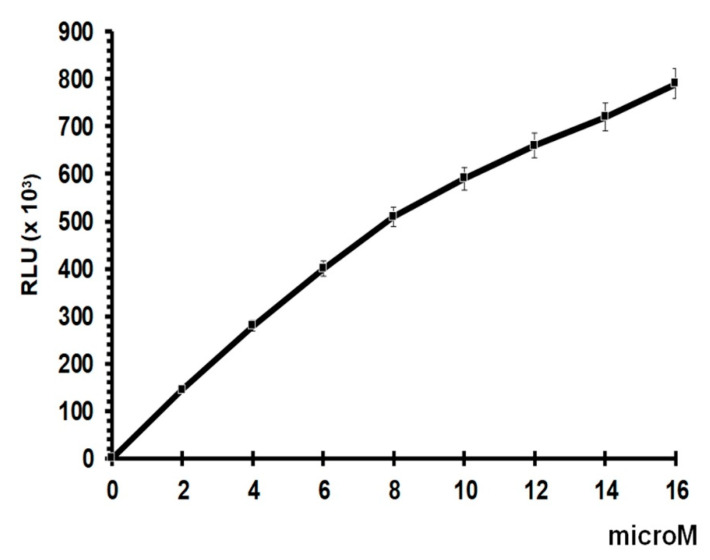
Glutathione standard curve.

**Figure 3 antioxidants-09-01291-f003:**
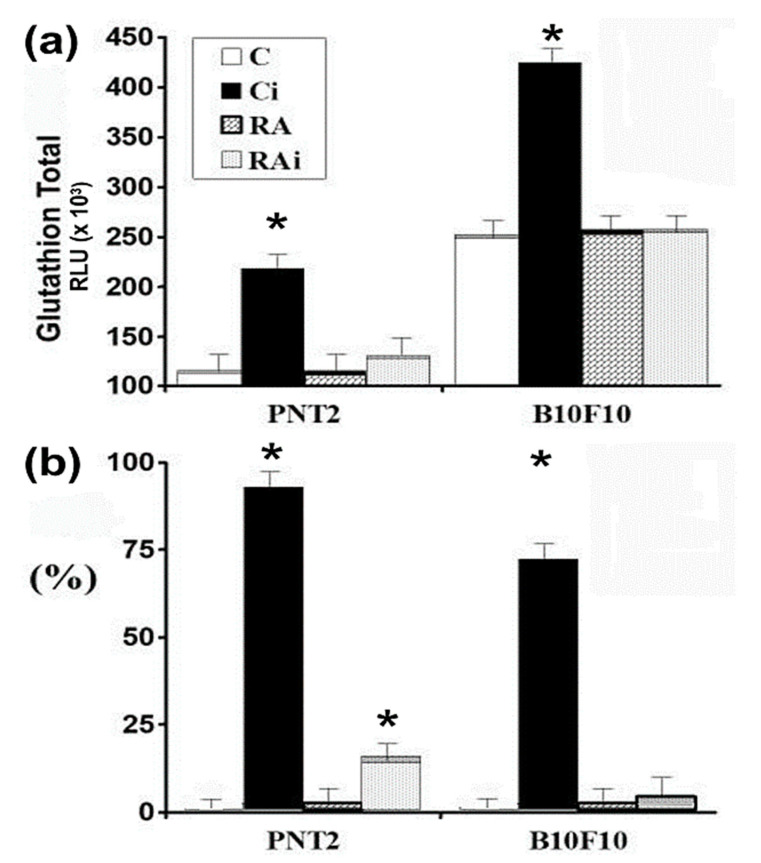
Total glutathione concentration in human prostate epithelial (PNT2) and B16F10 cells determined at 1h after exposure to 20 Gy of X-rays: (**a**) amount of total glutathione (GSH); (**b**) percentage increase with respect to control cultures (C—unirradiated controls, Ci—irradiated controls, RA—treated with RA dissolved in PBS, RAi—treated with RA dissolved in PBS and irradiated, * *p* < 0.001). Data are the mean ± standard error of eight independent experiments.

**Figure 4 antioxidants-09-01291-f004:**
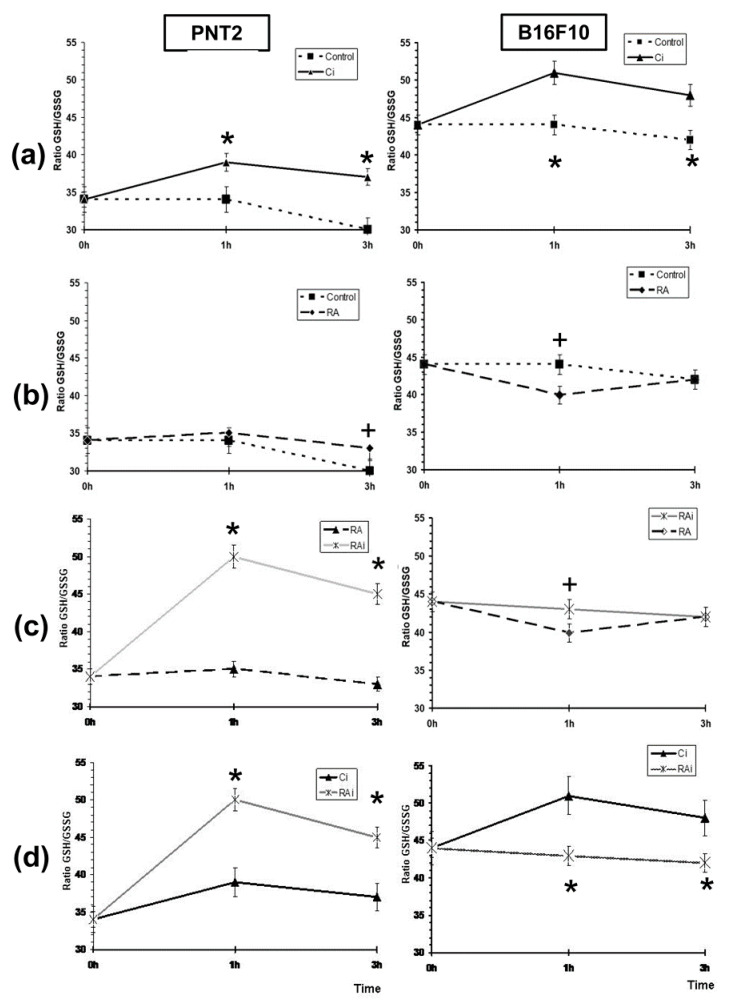
GSH/oxidized glutathione (GSSG) ratios in PNT2 and B16F10 cells determined at 1h and 3h after exposure to 20 Gy of X-rays: (**a**) GSH/GSSG ratio in unirradiated and in irradiated control cells; (**b**) GSH/GSSG ratio in unirradiated control cells and in RA-treated cells; (**c**) GSH/GSSG ratio in RA-treated cells and in irradiated RA-treated cells; (**d**) GSH/GSSG ratio in irradiated control cells and irradiated RA-treated cells (C—unirradiated controls, Ci—irradiated controls, RA—treated cells with RA dissolved in PBS, RAi—irradiated cells treated with RA dissolved in PBS, * *p* < 0.001, ^+^
*p* < 0.01). Data are the mean ± standard error of eight independent experiments.

**Figure 5 antioxidants-09-01291-f005:**
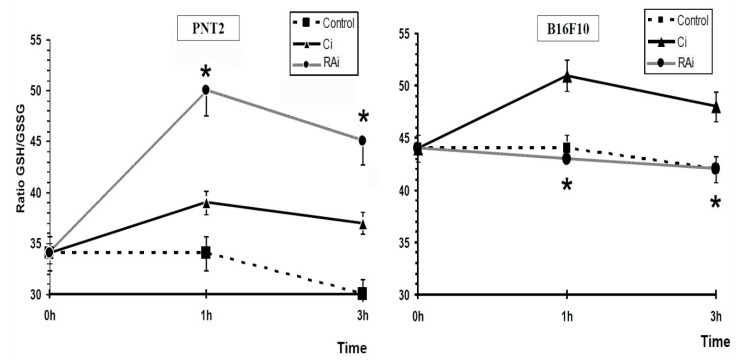
GSH/GSSG ratio in PNT2 and B16F10 cells determined at 1h and 3h after exposure to 20 Gy of X-rays (C—controls, Ci—irradiated controls, RAi—irradiated cells treated with RA dissolved in PBS, * *p* < 0.001). Data are the mean ± standard error of eight independent experiments.

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
