# Peer review of "Effect of Rosmarinic Acid and Ionizing Radiation on Glutathione in Melanoma B16F10 Cells: A Translational Opportunity"

_antioxidants, 2020, doi:10.3390/antiox9121291_

Round 1

Reviewer 1 Report

This study investigated the effect of rosmarinic acid (RA) on the melanoma B16F10 cells versus human prostate epithelial cells (PNT2) in the GSH intracellular production. This study has done a very simple study and observed the results of a simple phenomenon. The study for the intracellular mechanism on RA is also not investigated.
I have some comments and questions as follows:

1. You should study antioxidant defense via direct interaction with ROS or via activities of detoxication enzymes like SOD and glutathione peroxidases (GPx).

2. You should change English
Line 23, this type of substances --> this type of substance
Line 24, provide protection to --> protect
Line 36, radiosensitizing agent --> a radiosensitizing agent
Line 45, mechanisms affects --> mechanisms affect
Line 51, could provide protection to healthy --> could protect healthy
Line 58, Bovine serum albumen --> Bovine serum albumin
Line 94, was confirmed by means of thermoluminescent --> was confirmed utilizing thermoluminescent

Reviewer 2 Report

The present work by Olivares et al looks at the effects of Rosmarinic acid (RA) on an immortalized human prostate epithelial cell line (PNT2) and murine melanoma cells (B16F10) following exposure to high dose (20Gy) X-rays. The authors measure reduced glutathione at early timepoints (1, 3h) as a proxy for free radicals/oxidative stress. The authors argue that RA is protective in the immortalized cell line following high-dose irradiation which would argue for application of this drug concomitantly with irradiation of tumours to protect neighbouring, non-cancerous tissues.

The manuscript is well-written and to the point. Notwithstanding minor issues concerning presentation of the data/statistics, I do have a major specificity concern which prevents me from recommending publication unless it can be adequately addressed: 

In Reference 11 (Alvarez et al, 2014), the authors used the same cell models to address the effects of RA on cell viability following X-Ray irradiation. There was a large effect of DMSO on radiosensitivity (10Gy) in the same PNT2 cell line (Figure 4) which seemed to be radioprotective at 24 and 48 hr. In fact, DMSO appeared to be more radioprotective than RA. DMSO is a known free radical scavenger and as such is radio-protective in and of itself. In the present study it wasn't clear whether the authors used vehicle controls. Was RA dissolved in DMSO, and if so were controls treated with a similar volume of DMSO? If not, these experiments would need to be repeated to discount any effects of the DMSO vehicle on the radiation-mediated changes in glutathione levels. Experiments using another vehicle or an experiment showing a dose-response effect of RA are required.

Thus, the authors' assertion that RA has a specific, radioprotective effect cannot be adequately evaluated in the context of the current work as presented.

Reviewer 3 Report

This manuscript studies showed that the efficacy of rosmarinic acid is due to glutathione in cells. First, the author must explain the difference from the previous study such as below.

“ Berivan Tandogan et al., In vitro effects of rosmarinic acid on glutathione reductase and glucose 6-phosphate dehydrogenase. Pharm Biol. 2011 Jun;49(6):587-94. “

Overall, the following additional data is required:

  1. Ascorbic acid should be put into the data as a control group.
  2. I don't understand the need for Fig2 and meaning for Fig3b.
  3. Various analyses of anioxidants are required.

Ex) DPPH assay, Ferric reducing antioxidant power assay, Nitro blue terazolium reduction assay, ROS assay

  1. It is necessary to present mechanisms for rosmarinic acid glutathione and

Ex) Caroline Gaucher et al., Glutathione: Antioxidant Properties Dedicated to Nanotechnologies. Antioxidants (Basel). 2018 May; 7(5): 62.

Montserrat Marí et al, Mitochondrial Glutathione, a Key Survival Antioxidant. Antioxid Redox Signal. 2009 Nov; 11(11): 2685–2700.

Round 2

Reviewer 1 Report

It requires some proofreading of English, but appears to be of quality to be published.

Reviewer 2 Report

Thank you for clarifying which vehicle was used in the drug treatments and presenting the DMSO data which is quite interesting. However, this critical detail is still not mentioned in the revised version. If PBS was used as the vehicle for the 30uM RA treatments, this needs to be explicitly stated in the Methods and/or Results/figures. The "control" used in figures needs to be explicitly defined in the text/figures (i.e., "PBS" or "untreated"). 

Author Response

Comment.Thank you for clarifying which vehicle was used in the drug treatments and presenting the DMSO data which is quite interesting. However, this critical detail is still not mentioned in the revised version. If PBS was used as the vehicle for the 30uM RA treatments, this needs to be explicitly stated in the Methods and/or Results/figures. The "control" used in figures needs to be explicitly defined in the text/figures (i.e., "PBS" or "untreated").

Response. We have incorporated those changes. Please see lines 87, 145, 150, 151, 158, 165, 200 and 218.

We thank the referee for their contributions and the effort made that has allowed us to significantly improve the text presented.

Reviewer 3 Report

I understood the author's answer well. It will be a great help to understand the manuscrips if these contents are well reflected in the introduction or discussion. The results of the simple study in the future are believed to need more supplementation.